# Spatiotemporal Patterns and Influencing Factors of Industrial Ecological Efficiency in Northeast China

**Wai Li** 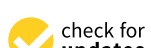**, Xiaohong Chen and Ying Wang \***

College of Geographical Science, Harbin Normal University, Harbin 150025, China
\* Correspondence: dilixiwangying@126.com; Tel.: +86-150-4542-4771

**Abstract:** Scientific measurement of regional industrial ecological efficiency and discussion of the development and changes of its spatiotemporal pattern are of great significance to accelerate the high-quality development of regional economy and coordinate the development of industrial economy and ecological environment. Taking the old industrial bases in Northeast China as the research case and 2004–2019 as the research period, a super-slack-based model was used to measure the industrial ecological efficiency of 34 prefecture-level cities in the region. Meanwhile, the spatial autocorrelation model and the geographically and temporally weighted regression (GTWR) model were used to analyze the spatiotemporal pattern characteristics and the spatiotemporal heterogeneity of influencing factors. The results showed that: (1) From a time change perspective, the overall industrial ecological efficiency of Northeast China declined, the mean of the 34 cities decreased from 0.675 to 0.612, the number of cities with a high level of industrial ecological efficiency decreased significantly, the number of cities with a low level of industrial ecological efficiency increased significantly, and the development gap between cities within the region widened. (2) In terms of spatial pattern, the difference in the spatial pattern in the east–west direction decreased, and the spatial pattern in the south–north direction was enhanced. The industrial ecological efficiency of the central part of Northeast China gradually became the highest in the whole region. (3) From 2017, the industrial ecological efficiency had stable spatial autocorrelation characteristics. The local spatial autocorrelation was dominated by L-H-type cluster areas in the mountainous regions and L-L-type cluster areas in central and southern Liaoning province. H-H and H-L types had small numbers. In addition, the trend of H-H cities transforming into H-L cities was obvious, and the high level of urban space spillover effect showed good development. (4) The science and technology input, industrial agglomeration intensity, and environmental regulation of the government generally had a promoting effect on the improvement in industrial ecological efficiency, while the economic extroverted degree had a negative impact. The high-value area of science and technology investment and industrial agglomeration intensity concentrated significantly in the central part. The government focused on ecological protection areas and ecologically sensitive areas, and the economic extroverted degree had a significant positive impact on the two major urban agglomerations in central Northeast China. Therefore, differentiating measures should be taken according to the actual situation of each city to improve the industrial ecological efficiency level in Northeast China.

**Keywords:** industrial ecological efficiency; GTWR; Northeast China; Super-SBM model

## 1. Introduction

Industrial ecological efficiency is characterized by the ratio of industrial output value to environmental pressure caused. The emphasis is on achieving the maximum industrial economic output with the minimum input of resources and environment and the minimum pollution emissions [1]. Industrial ecological efficiency can effectively measure the balanced relationship between economic development, environmental protection, and resource conservation. Exploring industrial ecological efficiency is of great significance to change the industrial development model and achieve sustainable development [2–5]. Early

academic research on industrial ecological efficiency mainly focused on the discussion of measurement methods of industrial ecological efficiency by Quariguasi, HOH, Kuosmanen, Andrés, and others [6–9]. Empirical research was carried out in the micro fields, for example, the analysis by Ma, S.J., Stergiou, Daria, and Patricia et al. on the industrial ecological efficiency of enterprises [10–14]. However, only from the micro perspective, the research could not effectively guide the reasonable development of industrial activities on the macro scale. Therefore, in recent years, the research perspective of industrial eco-efficiency has been shifted to the macro-regional level. Geographical disciplines that are good at regional and spatial analysis have more common research results in this field.

The regional-scale research on industrial ecological efficiency carried out by geographers was mainly reflected in three aspects. First, a study of the difference in industrial ecological efficiency inside the region was conducted. Many scholars, such as Tang, Z.L., Shao, L.G., Yu, X., Zhang, J.X., and Guo, studied the temporal change and spatial heterogeneity of industrial ecological efficiency in countries, urban agglomerations, and provinces, aiming at providing a decision-making reference for narrowing the gap within the region and comprehensively improving the industrial ecological efficiency [15–19]. Second was the space spillover effect of industrial ecological efficiency. As the relationship between geographical parameters was inversely proportional to the geographical distance, the lesser the distance, the stronger the relationship was. Xiao Qinlin, Liu Jia, Zhang Han, Tong Yun, and other scholars believed that the distribution of industrial ecological efficiency also accorded with this rule [20–23]; therefore, they used the spatial autocorrelation and spatial Dupin model to verify the spatial connection and correlation of industrial ecological efficiency. Third was the study of the influencing factors for the spatiotemporal differentiation of industrial ecological efficiency. Many scholars had carried out relevant studies, such as Shi, Y., Zhang Xinlin, and Lu Chengpeng et al. [24–30]. It mainly uses relevant models to discuss the strength of factors such as scientific and technological research and development, economic development level, industrial agglomeration, environmental regulation, industrial structure, and so on.

However, the existing regional-scale research results still have two deficiencies. First, the differentiation of natural and human environments determines the complexity and diversity of industrial regions. The pattern and process of industrial ecological efficiency evolution in different regions are the scientific bases for the harmonious development of man and land. The existing research is mainly conducted on the spatial scale of national, urban agglomeration, or administrative regions. No in-depth discussion is carried out based on different industrial regions. Second, the existing in-depth research mainly focuses on the evaluation methods of industrial ecological efficiency itself. The analysis of influencing factors for the spatiotemporal differentiation characteristics of industrial ecological efficiency is too simple. Scholars have conducted only a preliminary regression analysis of the selected influencing factors. No spatial heterogeneity of the influencing factors is involved. This is not conducive to the in-depth discussion of the formation mechanism and more targeted management countermeasures.

China is in the stage of rapid industrialization and ecological civilization development under the national strategic leadership. The urgency of realizing the improvement in industrial ecological efficiency makes it the main battlefield of the current research on industrial ecological efficiency on a macro-regional scale. Among the many types of industrial regions in China, Northeast China is a traditional old industrial base, with its typical people–land relationship, resource endowment, and industrial development [31]. This study followed the research paradigm of geography research mainly analyzing the spatiotemporal pattern of industrial ecological efficiency and its causes, choosing Northeast China as a research case and using a super-slack-based model (SBM) to measure the level of industrial ecological efficiency of 34 cities in Northeast China. On this basis, the study explored the development and evolution process of the spatiotemporal pattern of industrial ecological efficiency and used the geographically and temporally weighted regression (GTWR) model to further analyze the spatiotemporal heterogeneity of the factors

influencing the evolution of the spatiotemporal pattern. The study aimed to provide the research results of industrial ecological efficiency in a special type of industrial area (old industrial base) and was expected to provide the basis for the mode selection and path design for accelerating the high-quality development of such a regional economy and coordinating the development of industrial economy and ecological environment.

## 2. Research Area and Research Method

### 2.1. Study Area Selection

The industrial development of the old industrial bases often goes through three stages: development, decline, and revitalization. Generally speaking, during the development period, the heavy chemical industry system based on abundant regional resources is ecologically inefficient and discharges large quantities of pollutants into the geographical environment. In the decline stage of resource exhaustion, industrial decline, and increasing environmental pollution, renovation and revitalization are imperative, and the industrial activities in the region have to be reorganized. Improving industrial ecological efficiency to realize regional sustainable development is a key problem in the revitalization stage.

Northeast China is a natural geographical unit with complete ecological types and structures and a relatively complete regional economic unit. It has jurisdiction over Liaoning, Jilin, and Heilongjiang provinces and 34 prefecture-level city administrative units with a total area of 787,300 km$^2$ (Figure 1). During the period of planned economy, Northeast China has formed an industrial structure with large and medium-sized state-owned enterprises as the main body and the production of machinery, energy, and raw materials. Industrial production has obvious characteristics of "high consumption, high emission, and high pollution." After the reform and opening up, the economy of Northeast China fell into a structural crisis, and the northeast revitalization strategy implemented in 2003 promoted the regional economy to move into a short "golden decade" of transformation and revitalization. In 2014, the structural economic problems in Northeast China were not fundamentally solved but once again encountered a low development trough. In 2015, the Chinese government officially launched the "second revitalization" of Northeast China.

In the present stage of China's efforts to build an ecological civilization and achieve high-quality development, it is a major task for Northeast China, which has entered the stage of "secondary revitalization," to transform the mode of economic development, optimize the economic structure, and enhance the driving force for growth. Under this background, it is of great theoretical and practical significance to scientifically analyze the industrial ecological efficiency of 34 cities in Northeast China since the revitalization of Northeast China, explore its temporal and spatial regularity, and clarify its main influencing factors, so as to promote the coordinated development of resources, environment, and social economy in Northeast China.

### 2.2. Research Data Indicators

#### 2.2.1. Industrial Ecological Efficiency Measurement Index

Based on the concept of industrial ecological efficiency, referring to the existing research results on the calculation of industrial ecological efficiency [25–30], combined with the availability of data and consulting relevant experts, this study selected eight indicators from two aspects of input and output to construct the index system of industrial ecological efficiency in Northeast China (Table 1). The input indicators included environmental investment, resource input, human input, and capital investment. The environmental input was represented by industrial wastewater emissions, industrial soot emissions, and industrial sulfur dioxide emissions, and the resource input was represented by industrial water and electricity consumption. The industrial practitioners and the industrial fixed-asset investment were chosen to represent human input and capital input. The output index was expressed by industrial economic production, specifically by the industrial output value. The research period of this paper is from 2004 to 2019, and the relevant data were mainly derived from the *China Urban Statistical Yearbook* of 2005–2020, and some missing

indicators were supplemented from the statistical yearbooks of the corresponding years of each province.

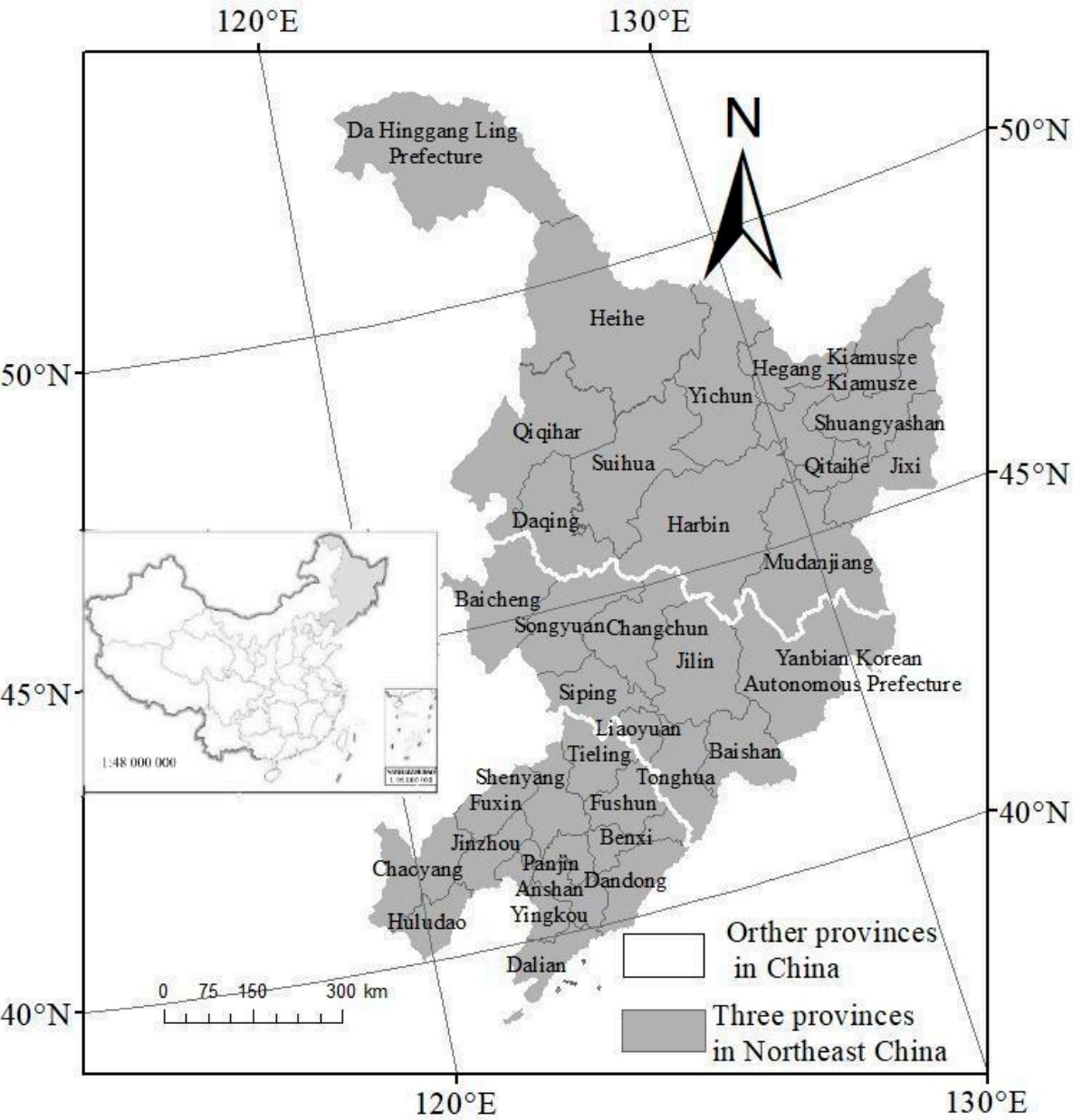

**Figure 1.** Schematic diagram of study area.

**Table 1.** Evaluation index system of industrial ecological efficiency.

| Level 1 Indicators | Level 2 Indicators | Level 3 Indicators |
|---|---|---|
| Investment index | Environmental input | Discharge of industrial wastewater (10,000 t) |
| | | Industrial soot emission (t) |
| | | Industrial sulfur dioxide emissions (t) |
| | Funding | Industrial fixed assets (CNY 10,000) |
| | Manpower input | Industrial employees (10,000 people) |
| | Resource input | Industrial water use (10,000 t) |
| | | Industrial electricity consumption (10,000 kW h) |
| Output index | Economic output | Industrial GDP (CNY 10,000) |

2.2.2. Analysis Index of the Driving Mechanism of the Spatiotemporal Evolution

Considering the characteristics of social and economic development in Northeast China and data accessibility in general, this study focused on the influence of the economic development level (pgdp), industrial agglomeration intensity (aggl), science and technology investment (tec), economic extroverted degree (open), and government governance (er) on industrial ecological efficiency. The specific proxy variables and measurement methods are shown in Table 2.

**Table 2.** Index of influencing factors for industrial ecological efficiency.

| Influencing Factor | Agent Variable | Measurement Method |
|---|---|---|
| Economic development level | Per capita GDP | Organized from *China Urban Statistical Yearbook* |
| Industrial agglomeration intensity | Location quotient | Proportion of regional industrial output in regional GDP/the proportion of national industrial output in national GDP |
| Science and technology input | Proportion of the expenditure on science, technology, and education | Science, technology, and education expenditure/GDP |
| Economic extroverted degree | Economic extroversion | Actual utilized foreign capital/GDP |
| Government governance | Industry $SO_2$ processing rate | Industry $SO_2$ production volume/industrial $SO_2$ handling capacity |

The influence mechanism of each factor on industrial ecological efficiency is as follows: (1) Economic development level: This impact is the most direct. According to the Environmental Kuznets Curve theory [32,33], natural resource consumption and pollution emissions intensify in the early period of rapid economic growth. However, with the continuous improvement in the economic development level, the industrial structure upgrading and technological progress lead to resource consumption and pollution emissions over the peak and then downward. (2) Industrial agglomeration intensity: The positive externalities formed by industrial agglomeration can improve the technical level and reduce the cost, which is conducive to intensive resource utilization and reduction of pollution emissions. However, the large agglomeration scale may also lead to the agglomeration diseconomy due to the "crowding effect," thus reducing the regional industrial ecological efficiency. (3) Science and technology investment: The improvement in innovation ability is an important guarantee for industrial green development. New production processes and technologies can significantly improve the level of industrial ecological efficiency. (4) Economic extroverted degree: External contact is a "double-edged sword" for regions. Foreign enterprises bring advanced technology and production process to reduce regional resource consumption and environmental pollution. However, foreign enterprises also adopt the regional loose environmental policy as the direction and treat the investment place as a pollution refuge. (5) Government governance: Increasing government governance can alleviate pollution and stimulate the innovative behavior of enterprises, thus improving environmental and economic performance and industrial ecological efficiency. The aforementioned index data of influencing factors were mainly compiled from the *China City Statistical Yearbook* during 2005–2020, thus forming a nonbalanced short-panel data set at the scale of prefecture-level cities in Northeast China from 2004 to 2019.

*2.3. Research Methods*

2.3.1. Super-SBM Model

Data envelopment analysis (DEA) is a new field of cross-study of operational research, management science, and mathematical economics. It is a quantitative analysis method to evaluate the relative effectiveness of comparable and similar units using linear planning methods based on multiple input indicators and multiple output indicators. The DEA

method and its model were established in 1978 by the famous American operations researchers A. Charnes and W. W. Cooper. The earliest DEA models were the CCR model with invariable scale remuneration and BCC models with variable scale remuneration [34,35].

In 2001, Kaoru Tone proposed a nonradial and no-angle SBM that was based on relaxation variables, which directly added the relaxation vector to the target function so that the economic explanation of the SBM model was to maximize the actual profit, not just the benefit ratio. On this basis, Tone further proposed the ultra-efficiency SBM model (Super-SBM) to solve the discrimination and sorting problem when the efficiency value of multiple decision units was one. The Super-SBM model overcame the defects of the traditional models. On the one hand, it effectively solved the relaxation problem of input–output variables, and on the other hand, it effectively solved the distinction and sorting problem when multiple decision units were effective at the same time. Therefore, the Super-SBM model could more veritably reflect the production efficiency than the other DEA models [36–39]. Accordingly, the Super-SBM model was used to measure the industrial ecological efficiency level in Northeast China as follows:

$$\min\rho = \frac{1 + \frac{1}{m}\sum\limits_{i=1}^{m} s_i^- / x_{ik}}{1 - \frac{1}{s}\sum\limits_{i=1}^{s} s_r^+ / y_{rk}} \tag{1}$$

$$s.t. \sum\limits_{j=1,j\neq k}^{n} x_{ij}\lambda_j - s_i^- \leq x_{ik}(i = 1, 2, \ldots, m) \tag{2}$$

$$\sum\limits_{j=1,j\neq k}^{n} y_{rj}\lambda_j + s_r^+ \geq y_{rk}(r = 1, 2, \cdots, s) \tag{3}$$

$$\lambda_j \geq 0, j = 1, 2, \cdots, n(j \neq k), s_i^- \geq 0, s_r^+ \geq 0 \tag{4}$$

where $x$ and $y$ represent the input and output variables, respectively; $m$ and $s$ represent the number of input and output indicators of the decision unit, respectively; the $\lambda_j$ represents the weight of the reference set; the $s_i$ and $s + r$ represent the relaxation variables of input and output, respectively; and $\rho$ represents the relative efficiency value.

2.3.2. Spatial Autocorrelation

The exploratory spatial analysis method was introduced to further analyze the spatial correlation and difference degree of industrial ecological efficiency between each city and its adjacent cities in Northeast China. Spatial autocorrelation refers to the potential interdependence between the observed data of some variables within the same distribution area. This study used the global Moran's I index to measure the spatial correlation and spatial difference of the overall objects, which could reveal the overall spatial characteristics of industrial ecological efficiency; the local Moran's I index was used to explore the spatial pattern of evolution and the outlier aggregation of industrial ecological efficiency. The method was as follows:

$$I = \frac{\sum\limits_{i=1}^{n}\sum\limits_{j\neq i}^{n} W_{ij}Z_iZ_j}{\sigma^2 \sum\limits_{i=1}^{n}\sum\limits_{j\neq i}^{n} W_{ij}}, \left[Z_i = \frac{V_i - \overline{V}}{\sigma}, \overline{V} = \frac{1}{n}\sum\limits_{i=1}^{n}(V_i - \overline{V})^2\right] \tag{5}$$

$$\text{Local Moran's } I = Z_i\sum\limits_{i=1}^{n} W_{ij}Z_j \tag{6}$$

where $I$ is the global Moran's I index; $n$ is the number of samples; $Z_i$ is the normalized transformation of $V_i$; and $W_{ij}$ is the adjacent spatial weight matrix of city $i$ and city $j$; when city $i$ and city $j$ belong to proximity relationship, $W_{ij} = 1$; otherwise, it is 0. The global

Moran's I index ranges between [–1,1]. $I > 0$ indicates a positive correlation in space, $I < 0$ indicates a negative correlation, and $I = 0$ indicates that industrial ecological efficiency is randomly distributed among cities. In Formula (6), a positive value of local Moran's I indicates the spatial agglomeration of factors with the same type of attribute values, and a negative value indicates the spatial agglomeration of factors with different types of attribute values.

2.3.3. Geographically and Temporally Weighted Regression

As an extension of the geographically weighted regression (GWR) model, the GTWR model is a spatiotemporal nonstationary regression model; the core is to add the time factor to the spatial GWR model. The model requires the addition of spatiotemporal coordinates in the analysis to calculate the space-time weight matrix. Traditional GWR analysis does not introduce the time dimension. However, GTWR forms spatial position coordinates and time series coordinates based on the GWR model, which considers the influence of both space and time on the regression coefficient of each explanatory variable [40]. In the space-time coordinate system, the coordinate of the space-time position $i$ is $(u_i, v_i, t_i)$. The GTWR model expression is as follows:

$$Y_i = \alpha_0(u_i + v_i + t_i) + \sum_{j=1}^{m} \alpha_j(u_i + v_i + t_i)X_{ij} + \xi_i \tag{7}$$

where $Y_i$ is the value of the explained variable of the sample point $i$ ($i = 1,2,3 \dots , n$), $n$ is the number of sample points, $m$ is the number of explanatory variables, $t_i$ is the time coordinate of the $i$th sample point; $\alpha_0 (u_i, v_i, t_i)$ represents the spatiotemporal intercept term of the sample point $i$, $X_{ij}$ represents the $j$th explanatory variable value of the sample point $i$, $\alpha_j (u_i,v_i,t_i)$ represents the regression coefficient of the $j$th variable at sample point $i$, which is a function of spatiotemporal coordinates, and $\xi_i$ indicates the residuals. By introducing space-time three-dimensional coordinates into the model, GTWR can improve the accuracy of model fitting and analyze the influence of each explanatory variable on the dependent variable from the perspective of three-dimensional space-time, which has good explanatory power.

**3. Results of the Study**

*3.1. Evolution of the Spatiotemporal Patterns of Industrial Ecological Efficiency in Northeast China*

Time Sequence Characteristics of Industrial Ecological Efficiency

The calculation results showed that the overall change in industrial ecological efficiency in Northeast China was divided into two stages (Figure 2). The first stage (2004–2013) was a period of steady growth during which the mean value of industrial ecological efficiency increased in 34 cities in Northeast China from 0.675 to 0.979, indicating that the northeast revitalization strategy had an obvious effect on the improvement in the industrial ecological efficiency. The second stage (2014–2019) was a rapid decline period. With the influence of the northeast economic downturn in 2014, the mean value of industrial ecological efficiency of 34 cities rapidly declined from 0.892 to 0.612 in 2019. This implied that the original balance of industrial development and ecological construction was broken, and the effect of ecological environment factors on industrial production was further enhanced. The main reason for the significant decrease in the overall industrial ecological efficiency was that the northeast industrial structure was still single and greatly affected by market changes, coupled with relatively backward technology and inefficient management.

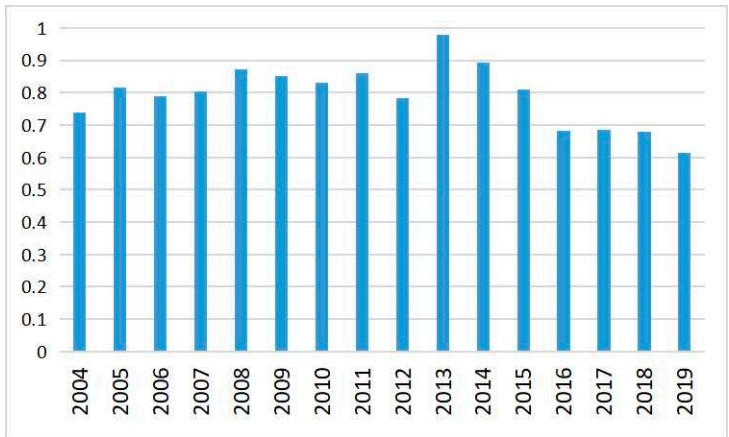

**Figure 2.** Change in the average value of industrial ecological efficiency in Northeast China.

The nuclear density curve can show the distribution of a group of data, which is the visualization of the "density" of a group of data on the coordinate axis. The density map is displayed by the fitted (smooth) curve. The higher the "peak" is, the more "dense" the data are, and the higher the "density" is. Draw the dynamic evolution trend of the nuclear density curve of industrial ecological efficiency in 2004, 2009, 2014, and 2019. Figure 3 shows that the center of the nuclear density curve of industrial ecological efficiency followed a trend of first right and then left, indicating that the concentration range of industrial ecological efficiency in 34 cities had an unstable change trend. In addition, the variation range of the curve showed increased volatility, indicating that the overall gap in industrial ecological efficiency had widened. In 2004, the curve presented double peaks, which was a pattern of "left main and right secondary". In 2009, the left peak moved right, and the right peak nearly disappeared, and the high value of industrial ecological efficiency decreased sharply. In 2014, the pattern of two peaks changed to "right dominant and left secondary" because the number of cities with high industrial ecological efficiency increased significantly, reflecting the positive influence of the northeast revitalization strategy. In 2019, a similar distribution pattern to 2004's was observed, but the left and right peaks decreased significantly. This change showed that the number of cities with industrial ecological efficiency above 1.0 in Northeast China gradually decreased, while the number of cities with low ecological efficiency increased, and the overall development momentum weakened. In this regard, we should strengthen regional cooperation, promote the transformation and upgrading of inefficient regional industrial structure, and further narrow the gap between cities.

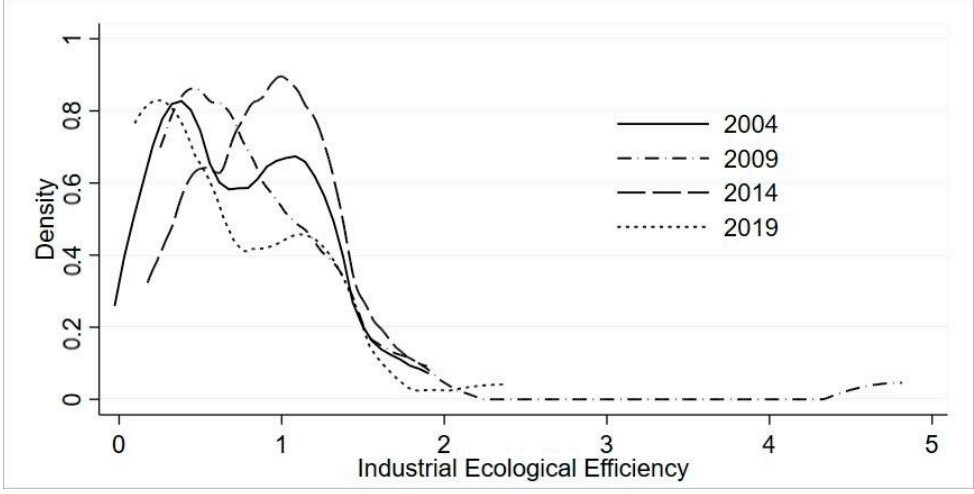

**Figure 3.** Dynamic evolution trend of industrial ecological efficiency in Northeast China.

With the support of ArcGIS 10.0 software, the fixed spacing method was used to render the spatial pattern of industrial ecological efficiency in Northeast China in 2004, 2009, 2014, and 2019 (Figure 4). As shown in Figure 3, in 2004, the spatial pattern of industrial ecological efficiency in Northeast China was mainly differentiated in the east–west direction. The industrial ecological efficiency value of the central and western cities was high, while that of the eastern cities was relatively low. Since then, the fluctuating and downward trend of industrial ecological efficiency in central and western cities has been obvious, and the development gap in the east–west direction has narrowed. At the same time, the spatial differentiation characteristics of "high in the middle and low in the periphery" in the south–north direction became increasingly prominent. From 2004 to 2019, the average industrial ecological efficiency of Liaoning, Jilin, and Heilongjiang provinces decreased from 0.77 to 0.36, 0.81 to 1.13, and 0.65 to 0.57, respectively. At present, the high-level industrial ecological efficiency in Northeast China shows a "block" distribution in northern Heilongjiang and central and western parts of Jilin province, and the low level shows a "block" distribution in central and southern Liaoning and eastern Heilongjiang province. These findings showed that the coal base in eastern Heilongjiang province and the heavy industry center in central and southern Liaoning provinces had low industrial ecological efficiency, which also meant that the distribution of industrial ecological efficiency might have a spatial correlation in Northeast China.

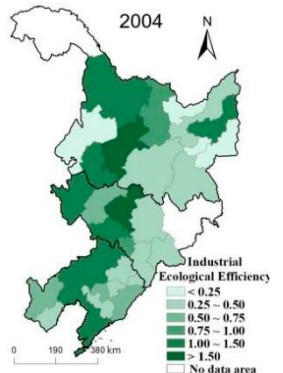 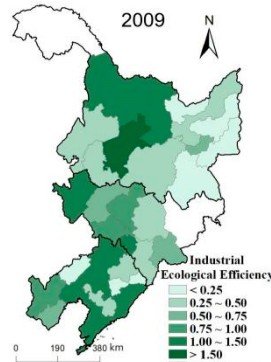 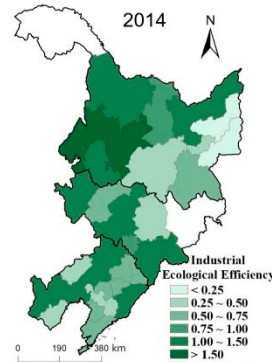 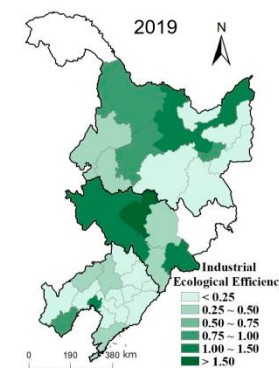

**Figure 4.** Evolution of the spatial pattern of industrial ecological efficiency in Northeast China.

The trend analysis tool was used to draw the spatial trend curve of industrial ecological efficiency (Figure 5), in which the *x*-axis is in the east and the y-axis is in the north. Figure 4 shows that from 2004 to 2019, the spatial pattern of industrial ecological efficiency in the 34 cities in Northeast China evolved from "high in the midwest and low in the east" to the inverted "U" structure of "high in the middle and low at both ends" in the east–west direction. At the same time, the curve arc decreased obviously, mainly due to the industrial ecological efficiency of all of the 34 cities having a downward trend; the decline was even greater in the central and western parts of Northeast China. In the north–south direction, the industrial ecological efficiency gradually evolved into an inverted "U" spatial pattern, and the north was slightly higher than that in the south. This change showed that the industrial ecological efficiency in the central region of Northeast China had a good development momentum. As shown in Figure 3, the central region of Jilin province centered on Changchun is the highest value distribution area of industrial ecological efficiency in Northeast China. This was mainly because Jilin province formed an industrial cluster centered on the automobile industry and constantly absorbed foreign capital and high-technology talents; thus, the industrial ecological efficiency was high.

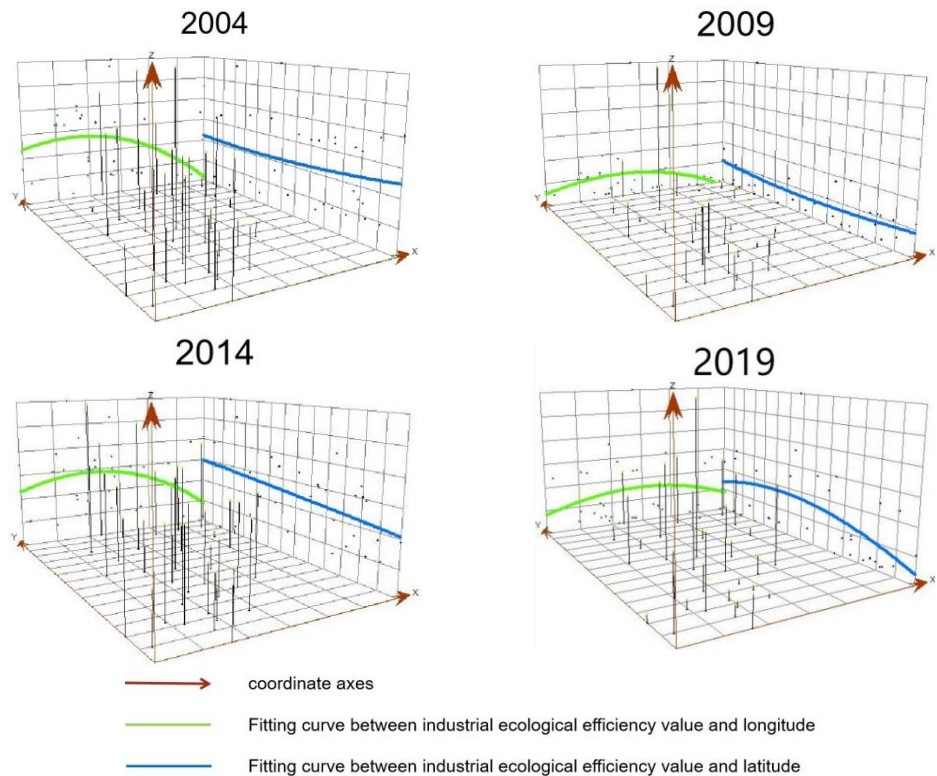

**Figure 5.** Spatial evolution trend diagram of industrial ecological efficiency in Northeast China.

### 3.2. Spatial Correlation Change in Industrial Ecological Efficiency in Northeast China

To verify the possible spatial correlation of industrial ecological efficiency changes among 34 cities, the global Moran index of industrial ecological efficiency in Northeast China from 2004 to 2019 was calculated using GeoDa software. Table 3 shows that the industrial ecological efficiency in Northeast China always fluctuated in the range of [–0.1,0.3] and was only significant at 10% in 5 years (2011, 2014, 2017, 2018, and 2019), indicating that the distribution of industrial ecological efficiency in space changed from strong randomness to spatial dependence. After 2017, the spatial autocorrelation characteristics of industrial ecological efficiency tended to be stable.

**Table 3.** Spatial autocorrelation results of industrial ecological efficiency in Northeast China.

| Year | Moran's | Z-Score | *p*-Value |
|------|---------|---------|-----------|
| 2003 | 0.075811 | 1.355522 | 0.175251 |
| 2004 | 0.055701 | 0.962019 | 0.336040 |
| 2005 | 0.101121 | 1.502341 | 0.133009 |
| 2006 | 0.109344 | 1.562383 | 0.118198 |
| 2007 | 0.104740 | 1.513525 | 0.130146 |
| 2008 | 0.092230 | 1.391958 | 0.163935 |
| 2009 | −0.026962 | 0.050239 | 0.959932 |
| 2010 | −0.007317 | 0.293490 | 0.769147 |
| 2011 | 0.151586 | 2.040306 | 0.041320 |
| 2012 | 0.037969 | 0.770232 | 0.441162 |
| 2013 | 0.024260 | 0.633757 | 0.526240 |
| 2014 | 0.289983 | 3.688333 | 0.000226 |
| 2015 | −0.000783 | 0.335593 | 0.737178 |
| 2016 | 0.015897 | 0.515338 | 0.606317 |
| 2017 | 0.193304 | 2.705608 | 0.006818 |
| 2018 | 0.218529 | 2.911985 | 0.003591 |
| 2019 | 0.155126 | 2.142546 | 0.032150 |

According to Formula (4), the scatter diagram of the local Moran index in 2017–2019 with stable spatial autocorrelation was drawn (Figure 6). According to the four quadrants of the scatter plot, the industrial ecological efficiency of 34 cities could be divided into four categories: the first quadrant was a high–high (H-H) cluster, which indicated that the industrial ecological levels of this city and adjacent cities were high, and the spatial correlation was at a high level. The second quadrant was a low–high (L-H) cluster area, which indicated that the industrial ecological efficiency of this city was low and that of adjacent cities was high, and the spatial correlation was at the development stage. The third quadrant was a low–low (L-L) cluster, which indicated that the industrial ecological efficiency of this city and adjacent cities was low, and the whole area was low efficiency. The fourth quadrant was a high–low (H-L) cluster area, which indicated that the industrial ecological efficiency of this city was high while that of adjacent cities was low, which was spatially manifested as a spillover effect.

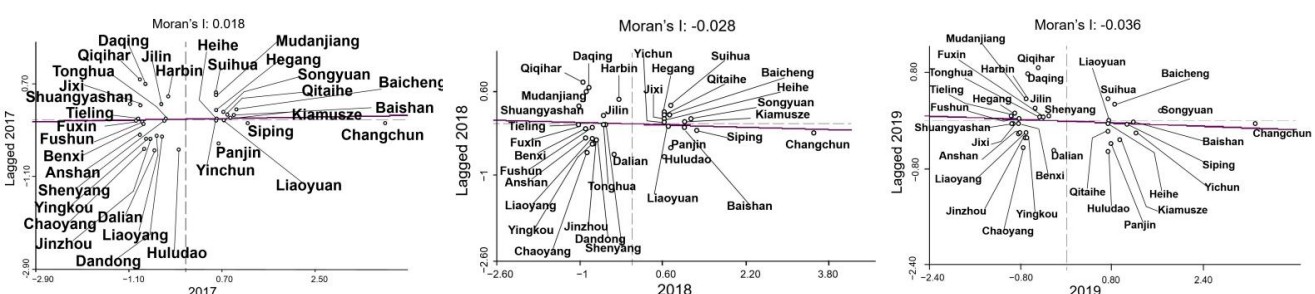

**Figure 6.** Local spatial autocorrelation of industrial ecological efficiency in Northeast China.

In 2017, the H-H cluster was found in northern Heilongjiang and western Jilin province. The L-H cluster was mainly distributed in the mountainous areas of eastern Northeast China and southwest Heilongjiang province, adjacent to the H-H cluster area. The H-L cluster had the least number of cities; only Changchun, Siping, Liaoyuan, Yichun, and Panjin showed a space spillover effect. In 2019, most of the cities in the H-H cluster in the north of Heilongjiang province changed to H-L type, and the number of cities in H-L clusters increased. However, Suihua, Baicheng, Songyuan, and other cities with high industrial efficiency had no obvious external impact, which was still H-H type. The H-H cluster was still dominated by the mountainous areas in eastern Northeast China and the southwest of Heilongjiang province. L-L-type cities such as Tieling, Fushun, and Fuxin in central and western Liaoning jumped to L-H-type cities. L-H cluster was still mainly in central and southern Liaoning, and resource cities such as Jixi and Shuangyashan in eastern Heilongjiang were added.

The proportion of cities with transition can reflect the local spatial stability of industrial ecological efficiency. The results showed that the agglomeration type of 18 cities—H-H-type Songyuan, Baicheng, and Suihua; L-H-type Harbin, Daqing, Qiqihar, and Jilin; L-L-type Dalian, Anshan, Dandong, Liaoyang, Jinzhou, Chaoyang, Yingkou, and Benxi; and H-L-type Changchun, Siping, and Panjin—did not change. A total of 16 cities completed the transition within 3 years, which indicated that the local spatial stability of industrial ecological efficiency was poor. In the future, high-efficiency areas should give full play to the spillover effect, avoid the formation of the "Matthew effect" dilemma, and promote the transformation of low-efficiency areas into high-efficiency areas. At the same time, low-efficiency areas should constantly improve their own industrial production system and actively accept the radiation and driving effect from high-efficiency areas.

### 3.3. Spatiotemporal Differentiation of Factors Influencing Industrial Ecological Efficiency in Northeast China

#### 3.3.1. Determination of the Main Influencing Factors

GTWR fitting was performed with the industrial ecological efficiency as the dependent variable and the influencing factors in Table 2 as the independent variables. The regression

fit R2 was 0.652; the fitting effect was good and passed the F test of 0.05. The fitting results show that the influence parameter of per capita GDP on the spatiotemporal pattern of industrial ecological efficiency was 0.000003, and the influence parameter of the proportion of the expenditure on science, technology, and education was 4.79. The influence degree of economic extroversion, industry $SO_2$ processing rate, and location quotient was basically the same; the three values were −0.03, 0.10, and 0.05, respectively. It showed that the economic development of Northeast China had a weak impact on the industrial ecological efficiency, and the promoting and inhibitory effects were not obvious, mainly due to the backward level of economic development. Under the background of the revitalization of the old industrial base in Northeast China, science and technology input became the most critical factor affecting the industrial ecological efficiency, which meant that the industrial ecological efficiency in Northeast China was obviously affected by the national strategy, which was also the main reason for the corresponding fluctuations in 2004–2019. The economic extroverted degree had a negative impact, which was in line with the "pollution shelter" hypothesis, indicating that the ecological environment threshold of foreign investment under the background of the old industrial base revitalization was less considered. The role of governance at this stage was not obvious, which was also related to the economic strength of the government. The comprehensive impact of industrial agglomeration intensity was positive, and the economy of scale and information technology exchange brought by agglomeration played a relatively obvious role in promoting industrial ecological efficiency.

### 3.3.2. Spatiotemporal Differentiation of the Main Influencing Factors

Science and education input: The northeast revitalization plan, which was compiled and implemented in 2003, focuses on improving the technological level and reducing the consumption of resources and pollution emissions. Therefore, the investment in science and education had the strongest impact on the improvement in the industrial ecological efficiency of cities in Northeast China from 2004 to 2019. The spatial differentiation pattern of its influence intensity has successively shown the change track of "high in the north, south and low in the middle," "high in the north and low in the south," and "high in the middle and east but low in northwest," which was an important reason for the transformation of the spatial pattern of industrial ecological efficiency in Northeast China in east–west and south–north directions (Figure 7). The northern part of Northeast China was not only a concentrated distribution area of resource-based cities but also an important national ecological barrier, resulting in more investment in the early stage of revitalization. In 2014, the overall economic downturn of Northeast China significantly reduced the corresponding investment, and the role of promoting industrial efficiency in the northern part of Northeast China was significantly weakened. The science and technology inputs of cities in central Northeast China were relatively weak in the early stage. However, as the bearing space of industrial revival, under the influence of ecological civilization and high-quality development strategy, these cities have carried out in-depth transformation of industrial production with their strong economic strength and scientific and technological level, which is also the main reason why the central region has become the highland of industrial ecological efficiency.

Economic extroverted degree: The impact of foreign direct investment on the industrial ecological efficiency of most cities in Northeast China was negative, and the interprovincial differences in the north–south direction were obvious. From 2004 to 2019, the spatial pattern of the action intensity of this factor changed from "Liaoning > Jilin > Heilongjiang" to "Jilin > Heilongjiang > Liaoning" (Figure 8). In terms of the internal differences in the three provinces, the high-value area of Heilongjiang province moved from the northeast to the central and western parts, Jilin province has been stable in the central part, and the high-value area of Liaoning province moved from the coastal to the central part. This was related to the scale and industrial field of foreign direct investment since the revitalization of Northeast China. Heilongjiang province is located at the forefront of the

China–Russia trade, but the urban economic scale and the foreign investment scale were small, and the industrial types were mainly agricultural product processing and related services. Therefore, the negative effects were relatively weak. The development level of the automobile industry cluster centered on Changchun in the middle of Jilin province is high, and the enterprises' absorption of advanced production technology was relatively obvious. The southern coast of Liaoning province was a concentrated distribution area of national and provincial economic development zones. However, the main foreign investment fields were machinery, medicine, metallurgy, and other industries, and the industrial competition among the cities was fierce, which caused obvious damage to the ecological environment.

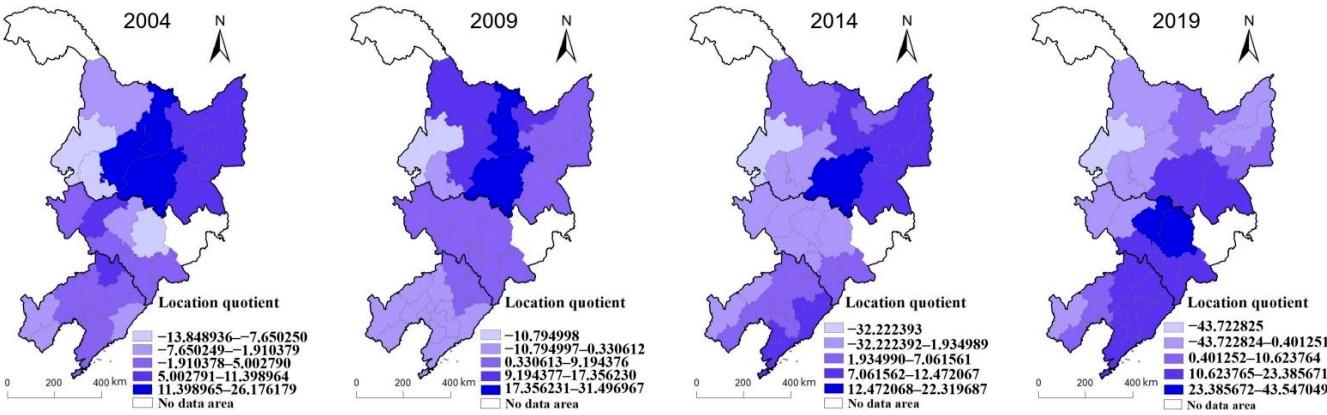

**Figure 7.** Spatiotemporal differentiation of the influence of science and education input.

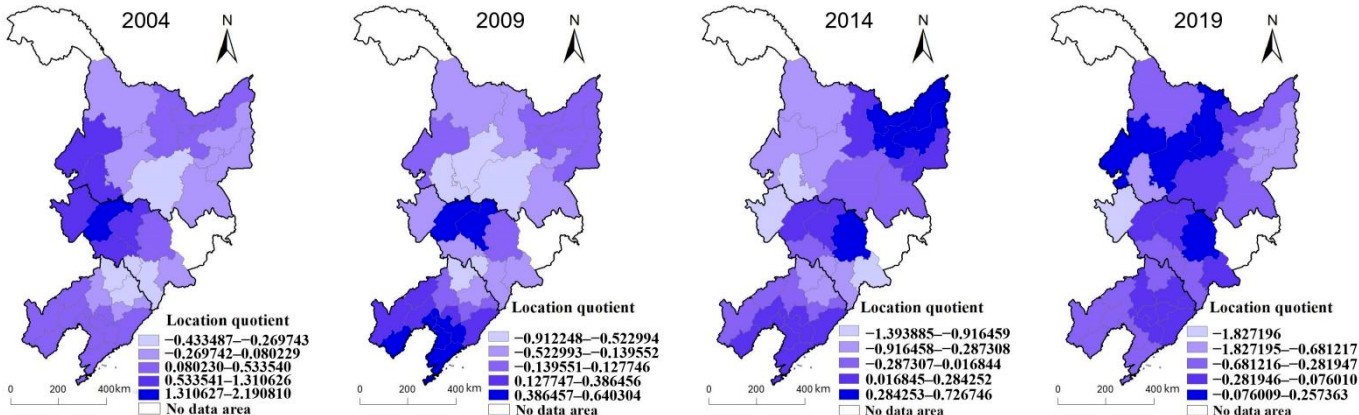

**Figure 8.** Spatiotemporal differentiation of the impact of economic extroversion degree.

Government governance: The government's pollution treatment capacity is an important embodiment of its macro-ecological environment development strategy. Its effect on the industrial ecological efficiency of cities in Northeast China was generally weak, and the spatial variation of the influence intensity was obvious. At the early stage, the influence of government governance in Heilongjiang and Jilin provinces was always strong. However, in 2009, the impact on Jilin and Heilongjiang provinces gradually weakened, while the impact on Liaoning province constantly increased (Figure 9). The areas where the government had a positive impact on industrial ecological efficiency were mainly distributed in the ecological barrier in northern Heilongjiang province, as well as the ecologically sensitive areas in western Jilin province and southern Liaoning province, which was also a concentrated distribution area with a high value of industrial ecological efficiency. However, effective environmental governance policies for the densely populated and industrially populated areas in central Northeast China were lacking.

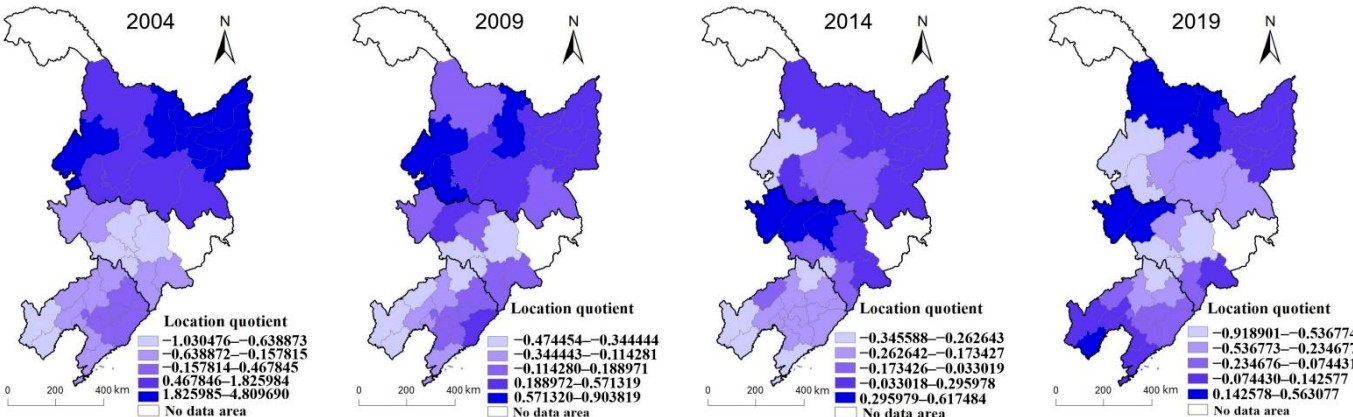

**Figure 9.** Spatiotemporal differentiation of the influence of government governance.

Industrial agglomeration intensity: From 2004 to 2019, the influence pattern of industrial agglomeration on the industrial ecological efficiency in Northeast China gradually changed from "high in the south and low in the north" to "high in the central and low in the periphery" (Figure 10). Urban agglomerations of Harbin–Chang and the central and southern parts of Liaoning province were key development areas. The improvement of the industrial agglomeration intensity made industrial enterprises produce positive externalities conducive to resource conservation and environmental friendliness through matching, sharing, and learning and promoted the improvement in industrial ecological efficiency. However, the industrial layout of each city in northern Northeast China was relatively scattered, the industrial scale was small, and a complete industrial cluster was not formed, which limited the improvement in industrial ecological efficiency.

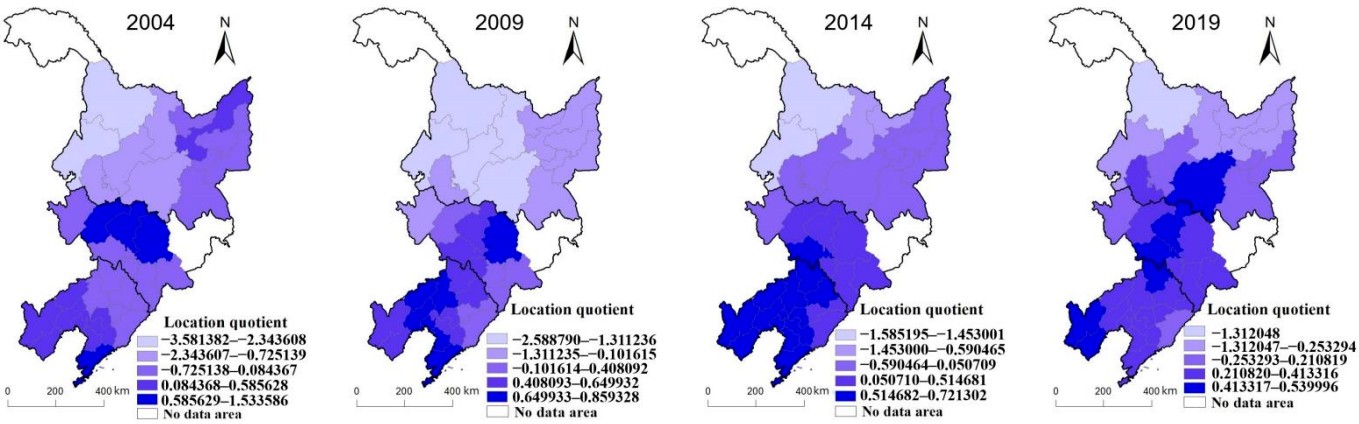

**Figure 10.** Spatiotemporal differentiation of the influence of industrial agglomeration on industrial ecological efficiency.

## 4. Discussion

(1). The analysis of the spatiotemporal differentiation characteristics of regional industrial ecological efficiency and its influencing factors is an important entry point for the current field of regional geography to pay attention to regional sustainable development. However, the existing research does not cover all types of industrial areas, and the analysis of influencing factors is relatively simple. This study selected Northeast China, which is in the revitalization stage, as the empirical research region. Based on the characteristics of its old industrial base, this study analyzed the overall change law of industrial ecological efficiency, the spatiotemporal differentiation characteristics, and the main influencing factors behind it in 34 prefecture-level cities. The research ideas and methods can provide a reference for other types of related research at the

regional level and also provide practical guidance for the improvement in industrial ecological efficiency and sustainable development in Northeast China.

(2). Based on the space-time pattern of industrial ecological efficiency and the space-time changes of influencing factors in Northeast China, this study had obvious policy implications. First, in the present stage, the government's environmental governance capacity in Northeast China is weak, and the impact on improving industrial ecological efficiency is not obvious. In the future, we should actively promote bilateral or multilateral government cooperation between internal cities, break the shackles of "information island" and "policy island," establish a long-term mechanism for regional collaborative governance, and gradually narrow the internal gap. Second, the impact of the economic extroverted degree and the government governance capacity on the industrial ecological efficiency in Northeast China is misplaced in space. This shows that, in the process of introducing foreign investment, Northeast China should reasonably plan the industrial structure, standardize the land-use mode, delimit the "restricted development" and "prohibited development" areas, improve the access audit mechanism of high-pollution and high-energy-consumption industries, and encourage and support the development of clean and sustainable industries. Third, Northeast China should formulate differentiated strategies according to different situations of internal industrial agglomeration. For the key industrial bearing space such as urban agglomerations of Harbin–Changchun and central and southern parts of Liaoning in the middle of Northeast China, the internal pattern of industrial agglomeration should be diluted, and the evolution from "single core" to "multi-core" pattern should be accelerated. This can promote the transformation from collective industrial agglomeration into professional industrial agglomeration, improve the relevance of agglomeration industries, and enhance regional competitiveness. For the northern, eastern, and western cities, it is necessary to focus on cultivating professional industrial agglomeration areas.

(3). This study was carried out in prefecture-level cities in Northeast China. However, the pattern and influencing factors of industrial ecological efficiency in Northeast China still need to be examined on more refined county and township scales. At the same time, this study selected only five influencing factors to analyze their influence on industrial ecological efficiency in Northeast China. Considering the actual situation, new influencing factors can also be added for the analysis. Moreover, the evolution of the spatiotemporal patterns of the industrial ecological efficiency still needs continuous observation and research with the promotion of the second revitalization of the old industrial base in Northeast China.

## 5. Conclusions

(1). The temporal variation in industrial ecological efficiency in Northeast China is characterized by low-level fluctuation and decline. From 2004 to 2019, Northeast China experienced the development process of "recession–revitalization–economic downturn–secondary revitalization". However, its industrial structure, dominated by traditional industries, still did not change significantly, and the industrial ecological efficiency also showed a fluctuating downward trend. Among the 34 cities in Northeast China, the number of cities with high-level industrial ecological efficiency decreased, and the number of cities with low-level industrial ecological efficiency increased. The pattern of development level among cities changed from "small gap at a high level" to "large gap at a low level".

(2). The spatial pattern of industrial ecological efficiency in Northeast China is mainly reflected in east–west and south–north differences. In 2004, the gap of industrial ecological efficiency in Northeast China was obvious in the east–west direction, with no significant difference in the north–south direction. In 2019, the spatial differences in the east–west direction decreased significantly, and that in the north–south direction expanded significantly. Specifically, the northern part of Heilongjiang province and

the central and western parts of Jilin province showed a "block" distribution of high industrial ecological efficiency, while the central and southern parts of Liaoning province and the eastern part of Heilongjiang province showed a "block" distribution of low industrial ecological efficiency. Since 2017, the industrial ecological efficiency of various cities in Northeast China began to have a stable spatial autocorrelation feature. L-H and L-L cluster areas had the largest number of cities. The former were mainly distributed in the eastern Heilongjiang province and the latter in central and southern Liaoning. H-H cluster areas were mainly located in northern Heilongjiang province and western Jilin province, while H-L cluster areas were mainly in the central Jilin province and the western coast of Liaoning.

(3). The spatial and temporal patterns of industrial ecological efficiency in Northeast China were affected by multiple factors, and the influence degree of each influencing factor for different cities in the region was also different. As the regional economic transformation was still in the exploratory stage, the impact of the economy on industrial ecological efficiency was not reflected. The level of science and technology input, government governance ability, and industrial agglomeration intensity had a positive effect on industrial ecological efficiency. Although the influence of economic extroverted degree was not strong, it showed a negative effect. Different factors had different effects on the industrial ecological efficiency of cities in Northeast China. The effect of science and technology input and industrial agglomeration intensity on improving ecological efficiency in central Northeast China was relatively obvious. The influence pattern of economic extroversion changed from "Liaoning > Jilin > Heilongjiang" to "Jilin > Heilongjiang > Liaoning." The government's environmental regulation focused more on reducing the environmental pollution caused by the industrial development of key ecological protection areas and ecologically fragile areas, such as the northern, western, and southern regions of Northeast China.

**Author Contributions:** Conceptualization, Y.W.; Data curation, W.L.; Project administration, X.C.; Software, W.L.; Visualization, W.L.; Writing–original draft, Y.W.; Writing—review & editing, Y.W. All authors have read and agreed to the published version of the manuscript.

**Funding:** This work was supported by the Natural Science Foundation of Heilongjiang Province (LH2019D008), the Innovative Youth Talent Cultivation Plan of Heilongjiang Provincial Universities (UNPYSCT-2018194), and the Youth Fund for Humanities and Social Sciences of the Ministry of Education (19YJC630177).

**Institutional Review Board Statement:** Not applicable.

**Informed Consent Statement:** Not applicable.

**Data Availability Statement:** Not applicable.

**Acknowledgments:** First of all, I would like to extend my sincere gratitude to my tutor, Wang Ying, for her guidance and contribution to this paper. Secondly, I would like to thank other authors for their help in the paper writing and other teachers in the school for teaching me knowledge and skills so that I had the ability to conduct research and write the paper. Finally, I would like to express my gratitude to all the students who provided me with help and suggestions in my paper writing.

**Conflicts of Interest:** The authors declare no conflict of interest.

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
