# Peer review of "Spatiotemporal Patterns and Influencing Factors of Industrial Ecological Efficiency in Northeast China"

_sustainability, doi:10.3390/su14159691_

Round 1

Reviewer 1 Report

Dear authors,

With great interest I read your manuscript, however the manuscript in its present form may need some improvement

1) Introduction: Citation needs improvement - it is not good enough to write 30 lines and than put 1-6 in parenthesis. The work of colleagues need to be identifiable and it is clearly not. One paragraph later it says 32-35. And what happend inbetween. This is not appropiate  citation practice and is a problem throughout the entire inroduction. Please download a recent sustainability manuscript for your orientation what readers expect.

2) The contextual relevance concerning the problem statement is missing. It is not good enough just to apply a specific methodology. Please indicate the reseach aim, goals and hypothesis.

3) Literature: A literature review related to your research case is completely missing. This needs to be added. Otherwise the reader can not follow the results and discussion and how they are put into perspective. 

4) Material and Methods: The authors indicate in L. 89 that this study is a research case. It needs to be outlined which approach to case study is followed and why this appropiate. So what school? How were cases selcted? What is the authors approach to rigor in line with their proceeding. Please add. This is important for the remainder of the methodological section

5) Results and discussion: Findings need to be put into perspective with the recent body of literature. This is missing

6) Why is there a discussion after the conclusion? For the conclusion can the authors critically reflect on their methods and work and add limitation. Can the authors make suggestion for future research

7) References: Can the authors check that intext citation as well as reference list are complete and correct and comply with the author guidelines.

Reviewer 2 Report

The abstract does not reflect the title. Too many variables that are not defined as part of the study are introduced. The objectives of the study should be clear.

The paper presents an “introduction” that ignore majority of the current literature. Thus, the gap is not grounded in science as well as in the literature.

The scope of a subject or paper do not necessarily reflect a search gap.

Eight indicators were selected to construct the industrial ecological efficiency index system in Northeast China How was this done?

The paper has no methodology. To what extent is the paper qualitative or quantitative in nature?

What are the research questions the study seeks to address and how are they connected to the methodology?

The authors discussed the area or scope of the study as methodology, that address the data collection approach. 

How does “2.2.1. Industrial ecological efficiency measurement index “ relate to the study is not clear

“The relevant data were mainly derived from the China Urban Statistical Yearbook of 2005–2020, and some missing indicators were supplemented from the statistical yearbooks of the corre- sponding years of each province This section of the paper is a repetition of past studies. The study is a repeat of the literature as indicated by the China Urban Statistical Yearbook of 2005–2020

“This study focused on the influence of the economic development level (pgdp), industrial agglomeration intensity (aggl), science and technology investment (tec), economic extroverted degree (open), and government governance (er) on industrial ecological efficiency” How does the  China Urban Statistical Yearbook of 2005–2020 relate to the statement?

How “Table 2Index of influencing factors for industrial ecological efficiency” was generated is not clear.

According to the Environmental Kuznets Curve (EKC) theory”. What is the Environmental Kuznets Curve  (EKC) theory and how does it relate to the present study?

The calculation results showed that the overall change in industrial ecological efficiency “ What is the nature of the change?

strategy had an obvious effect on the improvement in the industrial ecological efficiency. The second stage (2014– 2019) was a rapid decline period. With the influence of the northeast economic downturn in 2014, the industrial ecological efficiency rapidly declined from 0.892 to 0.612 in 2019” How this statement relates to the present study is not clear.

 With the support of ArcGIS 10.0 software, does this approach rely on  secondary data?  or compared to Data envelopment analysis (DEA)? 

Figs. 6-8 should be connected to the study for clarity.

The industrial ecological efficiency value of the central and western cities was high,  while that of the eastern cities was relatively low “ How does this relate to the title? How was this determined.

From 2004 to 2016, the average industrial ecological efficiency of Liaoning, Jilin, and Heilongjiang provinces decreased 289 from 0.77 to 0.36, 0.81 to 1.13, and 0.65 to 0.57, respectively”. How this was established is not clear.

The results are subjective in nature, as indicated in the analysis and discussion. Identifying the factors is not clear as in the current paper. The content of the paper does not reflect the title. Many and different variables are introduced at every stage of the paper. What the authors are seeking to communicate is not clear.

What are the influencing factors of Industrial Ecological Efficiency in Northeast China

The discussion section should be improved to indicate the gap in the literature. To what extent is the current study consistent with the literature?

What is the policy implication of the study?

How does the conclusion address the research questions?

The structure of the paper is not clear and does not support replication and knowledge transfer. How the study becomes a teaching tool or material is  affected for lack of clarity.

Reviewer 3 Report

How to improve the ecological efficiency of industry to realize the sustainable development of the region is a key problem not only for China, but also for other countries where traditional industry still plays an important role in the structure of the economy. Hence, I consider the research problem undertaken to be important. The article submitted for review requires some corrections before publication:

1) In the Abstract, the purpose of the research should be clearly formulated.

2) In the Introduction, the structure of the argument carried out should be briefly characterized.

3) In the Methodology, it should be explained why indicators such as Discharge of industrial wastewater (10,000 t); Industrial soot emissions (t); Industrial sulfur dioxide emissions (t) are treated by the authors as Environmental input, when emissions of all kinds are seen as a negative effect of human activity (economic activity).

4) The Discussion section should be moved before the Conclusions or merged with the Results section.

5) Why does line 506 have (3) at the beginning of the sentence?

6) The Conclusion section needs additions - it is intended to help the reader understand why your research should matter to them after they have finished reading the paper. A conclusion is not merely a summary of the main topics covered or a restatement of your research problem, but a synthesis of key points and, if applicable, where you recommend new areas for future research.

7) In my opinion, the References section was not fully prepared according to the Editor's guidelines.

Round 2

Reviewer 1 Report

The authors made serious efforts to improve the work and considered my and other reviewers comments.

Reviewer 2 Report

The authors have improved the quality of the paper. The comments have been addressed thus likely to improve international readership. However, sections of the results and analysis can be improved.

Reviewer 3 Report

Still, the References are not fully prepared according to the guidelines.

Author Response

This manuscript is a resubmission of an earlier submission. The following is a list of the peer review reports and author responses from that submission.